# Who cares about mental health? Benchmarking the issue importance of mental health for American voters"

Jake Haselswerdt*

Truman School of Government and Public Affairs, University of Missouri, Columbia, Missouri, United States of America
* haselswerdtj@missouri.edu

## Abstract

Existing scholarship on public opinion and mental health in the US has emphasized variations in Americans' stated support for policies or investments intended to address mental health. This work has shown that overall support for these policies is quite high, suggesting that scholars of public opinion may be focusing on the wrong dependent variable. This study asks a different question: to what degree is mental health an important voting issue for Americans, and what groups consider it especially important? Using a high-quality nationally representative sample of 1000 American adults from the 2024 Cooperative Election Study, I use recently developed experimental methods to assess how important hypothetical candidates' position on a mental health policy proposal (the Better Health Care for Americans Act) is to vote choice relative to nine other salient policy issues, including border security, abortion, and student loan forgiveness. The results suggest that mental health is of substantial importance, and especially so for liberals, higher-income people, and those in relatively poor health. These findings suggest that championing action on mental health could bring political rewards to policymakers.

## Introduction

How important is mental health to the American public as a policy issue? Mental health represents a growing public health crisis in the United States. Studies show that the incidence of mental health problems has grown in recent years [1], particularly during the COVID-19 pandemic [2], that many people with mental health problems struggle to gain access to treatment [3], and that the workforce of mental health providers is inadequate to meet the population's needs [4].

In recognition of these troubling patterns, public opinion scholars have sought to understand public opinion on mental health, with an eye toward understanding the politics of the problem and the feasibility of government action to address it. This includes observational studies on attitudes towards people with mental health problems as an outgroup [5], stigma [6] (including toward potential candidates for public

**Data availability statement:** The dataset and analysis code are available through Harvard Dataverse: https://doi.org/10.7910/DVN/ZPGCTP.

**Funding:** The survey module was paid for with internal funding from the Truman School of Government & Public Affairs at the University of Missouri. The funders had no role in study design, data collection and analysis, decision to publish, or preparation of the manuscript.

**Competing interests:** The authors have declared that no competing interests exist.

office with mental health problems [7]), personal experience [8], and partisanship [9], as well as experimental studies testing elements of messaging or framing [10]. This work has produced important insights on the politics of the issue and the likely effectiveness of different political strategies and messaging at building support.

As valuable as these studies are, however, their focus on variations in support or attitudes towards mental health risks losing sight of the forest for the trees. A good deal of evidence suggests that there is broad consensus in the American public that mental health is a serious problem for the country, and that government should take steps to address it. A 2022 Kaiser Family Foundation/CNN poll found that an eye-popping 90% of Americans agree that there is a mental health crisis in the U.S. [11], while a 2024 Gallup poll found that only 9% of Americans gave the American health system a grade of "A" or "B" for its performance on mental health, compared to 57% for "D" or "F" [12]. These numbers suggest that mental health is a "valence issue", in that the "goals or symbols" of improving mental health (if not the specific policy solutions) enjoy almost universal support [13]. While support does vary for specific policies, the general picture is one of decisive support. For example, Barry and McGinty [6] found 69% support for mental health insurance parity and 59% support for increased government spending, while Johnson and colleagues [14] found strong public support "for publicly funded mental health programs, both in terms of willingness to be taxed for program expansions and in terms of willingness to accept reductions in the number of beneficiaries in other public programs." Notably, the U.S. does not seem to be unique in this regard, as Bernardi finds that "The English public expresses strikingly high and stable over time levels of support for spending on mental health." [15].

While there is value in understanding why some people support government action on mental health while others do not, the apparently strong majority support for the former position suggests that other aspects of public opinion and political behavior may merit examination as well. Specifically, while Americans seem willing enough to express support for mental health policy solutions in costless survey responses, would they consider such issues important when making decisions in the voting booth? And what demographic or political groups place especially high or low importance on mental health as a policy issue? Existing theory and empirical findings in political behavior suggest that these questions are politically consequential: voters who see an issue as important are more likely to factor it into their choice of candidates, since important issues are more accessible or "top of mind" during decision-making [16,17].

To address these questions, I turn to the literature on issue importance in political behavior [18–20]. Studies in this literature focus not on voters' beliefs, but on the weight they put on some issues relative to others, typically in reference to their decisions between political candidates.

Using a recently developed method for assessing issue importance based on conjoint survey methodology [20] and a nationally representative sample of 1,000 American adults, I find that a candidate's position on a real mental health coverage and access proposal (The Better Mental Health Care for Americans Act of 2023) has

a significant and substantial effect on the reported likelihood of voting for the candidate, increasing the odds by 27 percentage points. This effect is at least comparable to those of more politicized and polarized policy issues (e.g., abortion access) and significantly greater than that of carbon emissions regulation. I go on to explore the political and demographic correlates of mental health issue importance, finding some suggestive evidence that higher-income people, ideological liberals, and those rating their own health as only "poor" to "good" place relatively high importance on mental health as a policy issue. There was no clear evidence that mental health issue importance varied across other demographic categories, including age, race, ethnicity, gender, education, or health insurance status.

This study contributes to the literature on public opinion on mental health by moving the discussion away from the question of reported support for action (which is very high) and toward the question of priorities. Political leaders are more likely to act on issues that affect voters' choice of candidates.

## Research design, data and methods

**Research design.** In this study I follow recent issue importance scholarship that uses random assignment of packages of issue positions in survey questions to measure the relative importance of issues for vote choice between two hypothetical candidates. Specifically, I adopt the approach of Alvarez and Morrier, who use dichotomous candidate issue positions (i.e., support or opposition for each policy proposal) in their experiment [20]. This approach presents survey respondents with relatively simple choices and is easily adaptable to existing and established survey questionnaires (see "Data and variables").

The intuition behind this experimental approach is best illustrated by an example. Imagine that a voter is confronted with a choice between candidate A and candidate B. The only information the voter has about these candidates is their positions on issues 1 and 2. The voter agrees with candidate A on issue 1 but not issue 2, and agrees with candidate B on issue 2 but not issue 1. The voter's choice between the candidates therefore reveals whether issue 1 or issue 2 is more important to them as a voter – for example, if they choose candidate A, they are willing to subordinate their disagreement with candidate A on issue 2 to their agreement on issue 1. By repeating this process with randomly assigned issues for each candidate pair and support-oppose positions for each candidate in a conjoint experiment, we can establish the relative importance of each issue to respondents' vote choice.

## Data and variables

I implement the conjoint experiment on a nationally representative module of 1,000 respondents on the 2024 Cooperative Election Study (CES), conducted in late 2024. [21]. First fielded in 2006 and run by the survey firm YouGov, the CES (formerly the Cooperative Congressional Election Study) recruits samples of American adults using a form of probability sampling called "sample matching" in which 60,000 members of existing online panels are matched to randomly selected citizens in 2023 American Community Survey microdata files on several demographic characteristics. YouGov randomly samples 1,000-respondent modules like the one used in the present study from the Common Content sample of 60,000. Survey weights are included to account for remaining discrepancies between the sample and the population, and I employ them in all analyses. More details on the theory, procedure, and accuracy of sample matching are included on pages 14–19 of the 2024 CES Guide [21].

I implement the conjoint experiment using a combination of original questions and items from the CES Common Content questionnaire. [21] An original question asks respondents if they support or oppose a specific mental health policy proposal based on the Better Mental Health Care for Americans Act of 2023: "Do you support or oppose the following proposal? Change health insurance rules and reimbursement rates to improve access to mental health care for all, including lower-income people, children, and the elderly." This format parallels a series of support/oppose items on the Common Content questionnaire (presented before the original module questions) that covers a wide range of issues. For the experiment, I incorporate responses to nine of those items, for a total of ten policy issues including mental health. The Common

Content issues include border security, abortion access, regulating carbon dioxide emissions, affordable housing, repealing the Affordable Care Act (ACA), student debt forgiveness, infrastructure investment, a "billionaire tax," and banning Tik-Tok. Complete wording of each policy proposal is displayed in the questionnaire in S1 Appendix [22]. While certainly not an exhaustive list of the universe of issues (let alone specific policy proposals), these issues vary widely in terms of their underlying subject matter. Their inclusion in the Common Content questionnaire is a testament to the fact that they are all politicized and in the public eye, making this a difficult test of the importance of mental health to vote choice.

With respondents' positions on the policy issues recorded, the survey proceeds to the conjoint experiment. I present respondents with six simple tables (each on a separate page) offering choices between pairs of generic candidates, described as Candidate A and Candidate B in each table. The only information provided about the candidates is their randomly assigned positions (support or oppose) on two policy proposals randomly selected from the list of ten. The only constraint on randomization is that the same policy proposal is not presented twice in the same table (though proposals do repeat across tables). Under each table, the respondent is asked to choose between candidate A and candidate B. An example question and table are displayed in the questionnaire in S1 Appendix [22].

The resulting dataset is at the candidate level, with the choice of candidate forming the dichotomous dependent variable (1 for choosing the candidate, 0 for choosing the other candidate). The nominal total sample size is therefore 12,000, but since each issue is analyzed separately, the effective sample size depends on the number of times the issue was randomly included in the tables, which ranges from 2,320–2,452. For each randomized candidate issue position, I created a dichotomous variable to indicate whether the candidate agrees with the respondent (1) or not (0). These indicators of respondent-candidate agreement are the key independent variables.

For the heterogeneous effects analysis, I make use of the Common Content's battery of demographic and political questions, collapsing some categories to achieve adequate subgroup sample sizes, an important concern for conditional analysis of conjoint experiments. [23] These include self-reported health (five-point scale collapsed to four categories), insurance status (indicators for employer, government, self-purchase, and uninsured), partisanship (three-point scale), ideology (three-point scale), education (six-point scale collapsed to four categories), family income (sixteen-point scale collapsed to three terciles), age (continuous measure collapsed to three terciles), race or ethnicity (indicators for identifying as White, Black, Hispanic, and other race, with the latter including Asian and all other categories), and gender (indicator for man). These variables are commonly used in studies of public opinion on mental health [8,9]. The wording of all Common Content questions is publicly available in the CES guide [21], and descriptive information on all variables is included in S2 Appendix [22].

The University of Missouri Institutional Review Board (IRB) found that this study posed no more than minimal risk to participants and declared it exempt from full IRB review on July 23, 2024. All CES respondents gave written informed consent to participate before beginning the survey, which took place between October 2 and November 2, 2024.

## Analysis

I follow the analytical strategy of Alvarez and Morrier [20] with a slight modification. Briefly, this approach estimates the average marginal component effect (AMCE) of respondent-candidate agreement on vote choice as the difference in the proportion or percentage of respondents who choose a candidate when they agree with the candidate on the issue compared to those who do not agree with the candidate on the issue. Since every element of the conjoint tables is randomized, there is no risk of bias from other elements (the other issue or the other candidate's positions). A more complete explanation of this strategy is provided in S3 Appendix [22]. I depart slightly from the Alvarez and Morrier approach to account for an item order effect: respondents were more than 12 percentage points more likely to choose "Candidate A" than "Candidate B" ($p < .001$). This is consistent with survey methodology work demonstrating that respondents in self-administered surveys tend to apply the "up (or first) means good" heuristic, evaluating items that are presented first (top or left) on the screen more positively, all else equal. [24] This introduces statistical noise that reduces precision.

Thus, I conduct OLS regressions (with survey weights and standard errors clustered by the respondent) with a control for candidate order. The coefficients for the agreement variables represent the AMCE estimates, which I present graphically with 95% confidence intervals. For the heterogeneous effects analyses, I run similar regressions incorporating interaction terms of the demographic and political variables described above with the mental health issue agreement indicator, using graphical analysis with 95% confidence intervals to assess which factors meaningfully condition the effect of issue importance. I operationalize all moderators, including ordinal measures, as factor variables, making these analyses flexible with respect to the functional form of any moderating relationships. I use Stata 19.5 SE for all analyses.

## Results

### Benchmarking the importance of mental health

Responses to the initial mental health policy question show widespread support, consistent with existing survey data: an estimated 91% support the proposal. Regardless of respondents' positions, however, the focus of this study is on how that position affects their reported vote choice.

Fig 1 displays the AMCEs for all ten policy proposals with 95% confidence intervals, with full results of the regressions displayed in S4 Appendix [22]. The estimated effect of agreeing with a candidate on the mental health proposal is large (27 percentage points) and statistically significant ($p < .001$). Mental health is roughly tied for third (with infrastructure spending and ACA repeal) out of this list of ten issues, behind only abortion access and student debt forgiveness. While the differences between the estimates for many issues are not statistically significant, mental health is significantly more important than regulating carbon emissions ($p = .03$ in a simultaneous estimation test). A cautious interpretation of these results suggests that mental health is of at least moderate importance to American voters even relative to highly salient and politicized issues, though more statistical power is needed to establish a clearer ranking.

Author's analysis of Cooperative Election Study 2024 module data. Graph shows average marginal component effects (AMCEs) and 95% confidence intervals calculated with separate ordinary least squares regressions for each policy issue

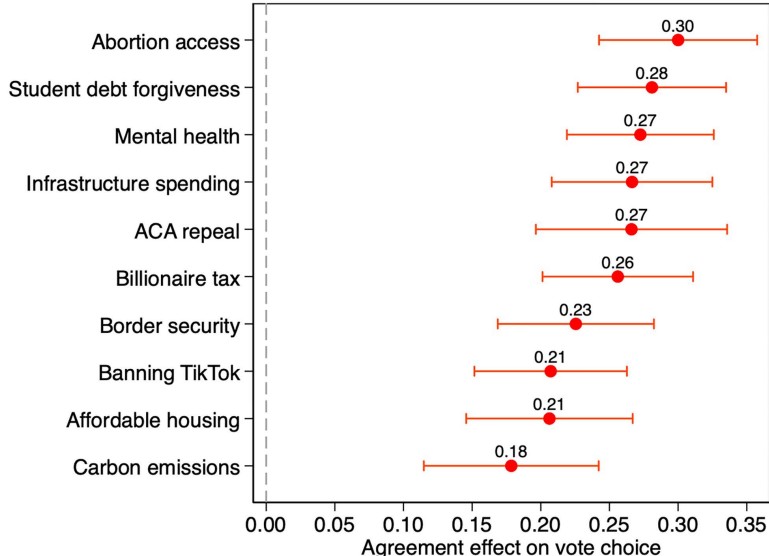

**Fig 1. Importance of mental health and other policy issues to reported vote choice (nationally representative survey of adults in the United States conducted in 2024).**

controlling for candidate position ("A" or "B") in the conjoint table. Analyses use survey weights and standard errors clustered by respondent. Full results tables are displayed in S4 Appendix [22].

## Subgroup differences in the importance of mental health to vote choice

Fig 2 summarizes the exploratory analyses of potential subgroup differences (i.e., heterogeneous effects). The plots display conditional effects (AMCEs) for each subgroup of each variable, derived from linear regressions incorporating interaction terms. It is important to acknowledge that the effective sample sizes in these analyses are smaller than is ideal for detecting conditional AMCEs in a conjoint study, per Schuessler and Freitag [23], and that the estimates presented are imprecise, especially for the smaller subgroups.

Author's analysis of Cooperative Election Study 2024 module data. Graph shows conditional average marginal component effects (AMCEs) and 95% confidence intervals calculated with separate ordinary least squares regressions for each policy issue controlling for candidate position ("A" or "B") in the conjoint table. Analyses use survey weights and standard errors clustered by respondent. Full results tables are displayed in S4 Appendix [22].

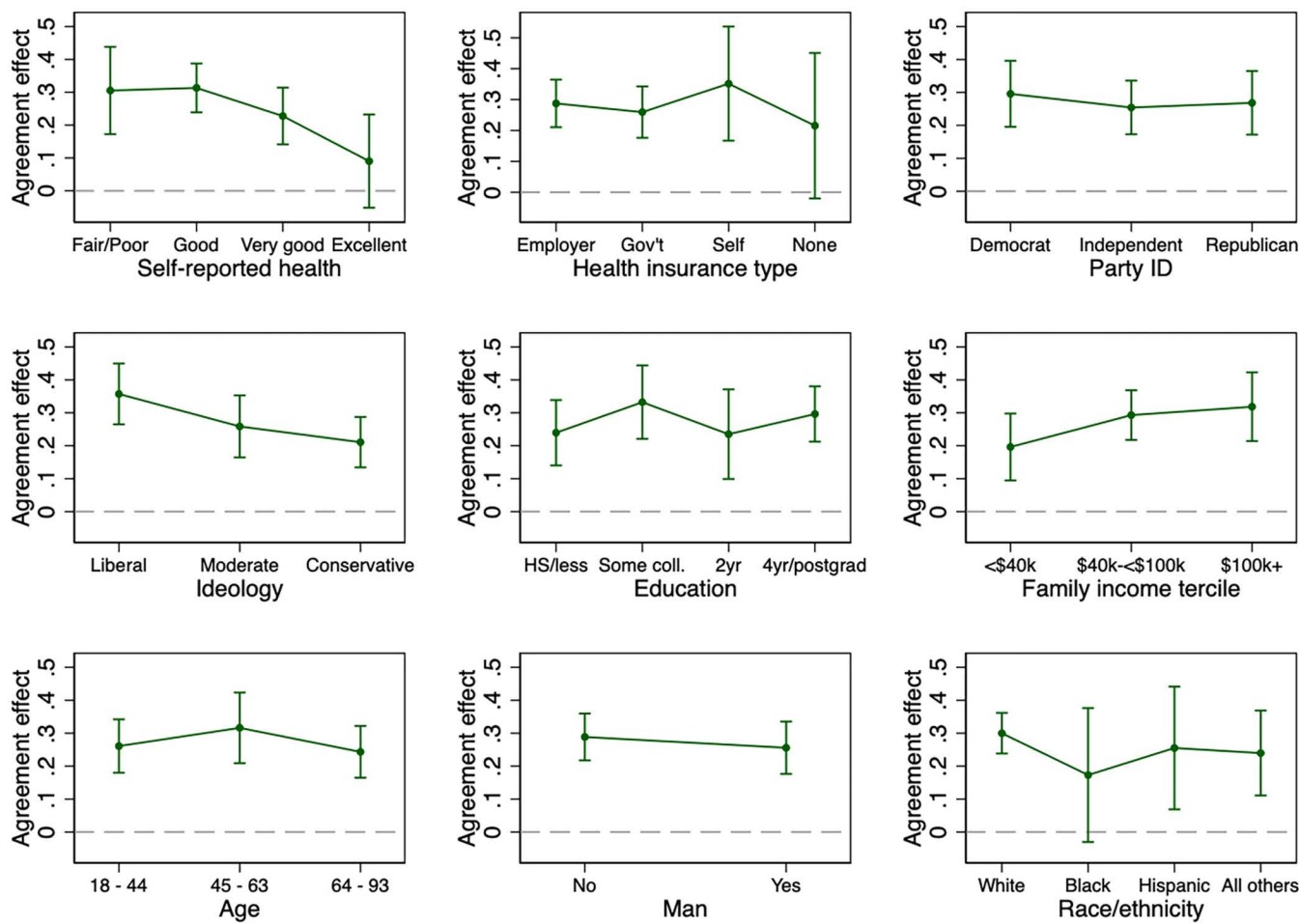

**Fig 2. Subgroup variations in the importance of candidate mental health position to vote choice (nationally representative survey of adults in the United States conducted in 2024).**

There is little clear evidence of heterogeneous effects, with a few notable exceptions: mental health appears to be substantially more important to those in poorer health compared to those in better health, to high-income people compared to low-income people, and to liberals rather than conservatives. While point estimates suggest the issue is more important for White voters than voters of color and more important for the self-insured than people in other health insurance categories, these estimates are not precise enough to reject the null hypothesis of no difference in effects across groups.

## Discussion and conclusion

This study represents a first attempt to bring together the study of public opinion on mental health with the study of issue importance and vote choice. The results demonstrate that a mental health coverage and access proposal is at least as important to Americans' reported vote choice as several much more salient and politicized issues, and especially important to certain subgroups: those in relatively poor health, liberals, and higher-income people. These findings add important context to the widespread support for government action on mental health documented in surveys, including this one. The present findings show that the public consensus in favor of policy action on mental health may carry real political weight. For politicians and policymakers, the findings suggest that championing mental health could promise political rewards at least comparable to some issues that are currently more prominent on the public agenda (e.g., border control), and possibly greater than some, especially regulating carbon emissions.

The apparent *importance* of mental health as a voting issue should not, however, be confused with its *salience* in the political sphere. Moniz and Wlezien argue that importance is a necessary but not sufficient condition for an issue to be salient – the latter implies that the issue is not only important, but the focus of attention [25]. A review of recent data on Americans' open-ended responses to the Gallup "most important problem" question suggest that mental health is not salient in this sense: it does not appear even among the issues identified by as low as 1% of the population as the most important problem [26]. Greater levels of attention from politicians and the media may be required for mental health to attain national salience.

Of course, this study suffers from important limitations. The most notable of these is the limited statistical power for a conjoint study [23], which prevents a clearer ranking of mental health's importance relative to most other issues. The relatively small sample size also results in imprecise estimates of subgroup effects for smaller groups (e.g., people of color), resulting in some ambiguous findings. Replicating this design with a larger sample is an obvious next step for this line of research.

Another limitation is common to all experiments involving policy proposals: the findings may be specific to the content or wording of the included proposals. In this case, the specific concern is that different proposals may evoke different issue frames, which research has shown may impact perceived issue importance [27]. While this is an important caveat, Alvarez and Morrier [20] explored this possibility by including four different climate change policy proposals in their study using this method, finding little difference in issue importance between them. This suggests that voters may consider the importance of issues in relatively broad thematic categories (e.g., "mental health" rather than a specific insurance parity regulation). Nevertheless, future research should consider a range of mental health policy proposals.

Another shortcoming is that the dichotomous nature of the policy position variables for both the respondents and the hypothetical candidates eliminates nuance. Different patterns could emerge if the analysis were able to account for degrees of agreement or disagreement. While this is not possible for the nine items on the Common Content (which included only "support" and "oppose" as response options), I am able to examine differences in agreement effects for the mental health proposal, since I included a question on strength of support or opposition on the module. The results of two such analyses (see Table S4.6 [22]) indicate that the agreement effect on vote choice is much higher for respondents who hold strong positions on the mental health proposal, and that this is apparently driven by the 74% of respondents who indicated strong support. Future work in this vein should consider the possibility that intensity of agreement or disagreement may affect vote choice differently for different issues.

## Supporting information

**S1 Appendix. Questionnaire.**
(DOCX)

**S2 Appendix. Descriptive Statistics.**
(DOCX)

**S3 Appendix. Conjoint Methodology.**
(DOCX)

**S4 Appendix. Full Regression Results.**
(DOCX)

## Acknowledgments

The author acknowledges Christopher Ojeda and other participants at the 2025 Health Politics and Policy Conference for their comments on this project.

## Author contributions

**Conceptualization:** Jake Haselswerdt.

**Data curation:** Jake Haselswerdt.

**Formal analysis:** Jake Haselswerdt.

**Investigation:** Jake Haselswerdt.

**Methodology:** Jake Haselswerdt.

**Project administration:** Jake Haselswerdt.

**Resources:** Jake Haselswerdt.

**Software:** Jake Haselswerdt.

**Supervision:** Jake Haselswerdt.

**Validation:** Jake Haselswerdt.

**Visualization:** Jake Haselswerdt.

**Writing – original draft:** Jake Haselswerdt.

**Writing – review & editing:** Jake Haselswerdt.

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
