## [Decision Letter · Decision Letter 0]

8 Dec 2025

PONE-D-25-40424Who Cares About Mental Health? Benchmarking the Issue Importance of Mental Health for American VotersPLOS One?

Dear Dr. Haselswerdt,

Thank you for submitting your manuscript to PLOS ONE. After careful consideration, we feel that it has merit but does not fully meet PLOS ONE’s publication criteria as it currently stands. Therefore, we invite you to submit a revised version of the manuscript that addresses the points raised during the review process.

We look forward to receiving your revised manuscript.

Kind regards,

Omar El Deeb

Academic Editor

PLOS One

**Journal Requirements:**

“The survey module was paid for with internal funding from the Truman School of Government & Public Affairs at the University of Missouri.”

“The author also acknowledges the Truman School of Government and Public Affairs for funding the CES module.”

“The survey module was paid for with internal funding from the Truman School of Government & Public Affairs at the University of Missouri.”

4. Please note that your Data Availability Statement is currently missing the DOI/accession number of each dataset OR a direct link to access each database. If your manuscript is accepted for publication, you will be asked to provide these details on a very short timeline. We therefore suggest that you provide this information now, though we will not hold up the peer review process if you are unable.

**Additional Editor Comments:**

Kindly take into account in your revised version all the comments raised by the reviewers and be keen to provide answers, refutations or suitable amendment in the manuscript. I would like the authors also to be aware about recent works related to voting behavior of American voters (https://doi.org/10.1142/S0129183124501900,
https://doi.org/10.1371/journal.pone.0331959) that may be relevant to their work, if they find informative and suitable.

The manuscript submitted is of interesting and original value, but needs to be revised according to all comments raised in the revision process.

Reviewers' comments:

Reviewer's Responses to Questions

**Comments to the Author**

1. Is the manuscript technically sound, and do the data support the conclusions?

Reviewer #1: Yes

Reviewer #2: Yes

Reviewer #3: Partly

2. Has the statistical analysis been performed appropriately and rigorously?

Reviewer #1: Yes

Reviewer #2: Yes

Reviewer #3: Yes

3. Have the authors made all data underlying the findings in their manuscript fully available?

Reviewer #1: No

Reviewer #2: Yes

Reviewer #3: Yes

4. Is the manuscript presented in an intelligible fashion and written in standard English?

Reviewer #1: Yes

Reviewer #2: Yes

Reviewer #3: No

Reviewer #1: Thank you for the opportunity to review this very interesting manuscript on public opinion towards mental health in the US and its implications for candidate support. Overall, I find the paper original and I believe it makes a nice contribution to the mental health and politics subfield. Below I have some comments for the author which I hope will help improve the paper.

It is interesting that mental health seems to be more important than other issues in 2024; although I would distinguish between support for and salience of the issue. I think the author can dig deeper here. When polling companies in the US like Gallup ask about the most important issue/problem (MII/MIP) in the country through an open-ended question, has mental health ever come up as one of the issues? Certainly, it was not the case in the past, but I wonder whether this has changed given that people have become more aware of the issue. If the author finds such evidence, then this can help with the external validity of the research. If the author, instead, does not find such evidence, then it is worth discussing the point anyway because when people are primed, they do think about mental health, as the paper shows. This is still a relevant point when thinking about priming effects and what polls should ask. Maybe they should start asking about mental health in MIPs, although support for a proposal does not necessarily mean that the issue is salient. There is research on public support for mental health outside of the US. For example, a paper that uses data from England (Bernardi 2021a) shows that support for mental health services is very high among the English public. So, this evidence is not unique to the US and acknowledging this can strengthen the message of the paper.

Mental health may be a valence issue (see Stokes 1992) – there is wide agreement on the goals at least, maybe not on the means –, that is, support may be high on mental health services and a relationship may be found with voting propensity. However, a totally different matter – in the US – is support for a candidate with mental health problems. The available experimental research (Loewen and Rheault 2021; Magni and Reynolds 2024) tells us that candidates with mental health problems suffer an electoral penalty. I believe such discussion should be incorporated in the paper. Mental illness stigma still is relevant in the US.

The paper dwells between mental health as a societal problem and as a public opinion’s priority. But the author does not engage at all with any of this research. As it stands, the paper is entirely empirical with no attempt to include any theoretical discussion or to connect the findings to existing research.

Why is there a positive effect of public approval of a mental health policy proposal (but also other proposals, based on Figure 1) and candidate support? For example, according to Kristensen et al. (2023: 2855), “If a problem indicator shows that a problem is severe, then parties across the

board attend to the problem.” In other words, candidates who advance policy proposals about pressing problems may be rewarded electorally because they are paying attention to such problems. Alternatively, as Fournier et al. (2003) suggest, when asked to evaluate the government on a certain issue when it is important to them, people are more likely to connect their evaluation to their vote choice. (For a broader discussion on issue salience I recommend the entry by Moniz and Wlezien (2020).)

And why is support for mental health so high in the US? Is it because mental health is a valence or quasi-valence issue (Bernardi 2021b)? In the UK, PhD research by Daniel Bowman (2025) shows that politicians’ attention to mental health has increased over time and the main parties (Conservative, Labour, Liberal Democrats) and parliamentarians from those parties all care about mental health to some extent but they frame the issue differently in election manifestos or parliamentary debates.

Overall, those are just some examples of literature from issue salience / agenda setting, but it would be nice if this paper could be a bit more theory-inspired and could provide some discussion of the results in that direction. The author may not have pre-registered hypotheses, but he/she should attempt to explain what is going on in their data.

Why are results presented with 90% and not 95% confidence intervals?

Of course, the experiment has been conducted, but I still wonder about the treatment condition for mental health. It is hard to disagree with such a statement, especially if one is only given the possibility to support or oppose the policy, without any opportunity to use a degree of options. In this regard, the author has asked about strength of support/opposition. How do results look like when a variable that incorporates those options is used? I think the author can construct an ordinal variable from the two original variables.

References:

Bernardi, L. (2021a). Mental Health and Political Representation: A Roadmap. Frontiers in Political Science, 2(January), 1–13. https://doi.org/10.3389/fpos.2020.587588

Bernardi, L. (2021b). Depression and political predispositions: Almost blue? Party Politics, 27(6), 1132–1143. https://doi.org/10.1177/1354068820930391

Bowman, D. (2025). Framing a Wicked Problem: Understanding the Changing Attention of UK Political Parties and Parliamentarians to Mental Health, 1997-2024. PhD dissertation. University of Liverpool.

Fournier, P., Blais, A., Nadeau, R., Gidengil, E., & Nevitte, N. (2003). Issue Importance and Performance Voting. Political Behavior, 25(1), 51–67.

Kristensen, T. A., Green-Pedersen, C., Mortensen, P. B., & Seeberg, H. B. (2023). Avoiding or engaging problems? Issue ownership, problem indicators, and party issue competition. Journal of European Public Policy, 30(12), 2854–2885. https://doi.org/10.1080/13501763.2022.2135754

Loewen, P. J., & Rheault, L. (2021). Voters Punish Politicians with Depression. British Journal of Political Science, 51(1), 427–436. https://doi.org/10.1017/S0007123419000127

Magni, G., & Reynolds, A. (2024). Voter attitudes and politicians with health conditions and disabilities. Social Science Quarterly, 105(6), 1923–1937. https://doi.org/10.1111/ssqu.13451

Moniz, P., & Wlezien, C. (2020). Issue Salience and Political Decisions. Oxford Research Encyclopedia of Politics. https://doi.org/10.1093/acrefore/9780190228637.013.1361

Stokes, D. E. (1992). Valence Politics. In D. Kavanagh (Ed.), Electoral Politics (pp. 141–164). Oxford University Press.

Reviewer #2: Thank you for the opportunity to review this interesting and well-executed study. The manuscript presents a timely and valuable contribution by shifting attention from general attitudes toward mental health policy—where public support is already high—to the question of whether mental health actually influences vote choice. The experimental design using a conjoint approach embedded in the CES is methodologically appropriate and well justified, and the findings are clear: mental health policy positions meaningfully affect respondents’ vote choices at levels comparable to other prominent issues. This offers an important addition to the literature on issue priorities in public opinion.

The analysis is rigorous overall, and the write-up is strong. I have several suggestions that could strengthen the manuscript further:

1. Subgroup analyses — The discussion would benefit from slightly more explicit emphasis on the imprecision of effects for smaller demographic groups. This is acknowledged, but a bit more caution in interpretation within the Results section would improve clarity.

2. Item-order effect — The detection of a significant preference for Candidate A and the analytic adjustment for this are important. Briefly elaborating on the theoretical or cognitive reasons such effects occur in conjoint tasks could help contextualize this issue for readers.

3. Generality of the mental health proposal — While the argument that voters respond to broad thematic categories is reasonable, a bit more discussion of how issue framing may influence relative salience would strengthen external validity claims.

4. Minor polish — There are a few formatting and typographical inconsistencies that can be easily corrected during revision (e.g., spacing and line break artifacts from the draft format).

Despite these modest suggestions, the study meets the technical and reporting criteria for PLOS ONE and adds meaningful evidence that mental health is not only symbolically supported but also politically consequential. I encourage the authors to continue developing this line of work.

Reviewer #3: Comment: Introduction:

Could you add more about the rationale? And a clear objective. Your objective has been spread all over the introduction. I understand what you intend to do, however, we need a clear and specific written objective.

Comment: Data and Variables : you said you sampled 1000 randomly, but how many where there in total? And did you do the randomization? If you didn’t opt for any randomization technique, then you should probably say that you sampled them using convenience sampling. Otherwise, explicitly mention how you randomized them.

Comment: in line 163-165, you said “implement the conjoint experiment using a combination of 164 original questions and items from the CES Common Content 165 questionnaire.”. Since you are referring to something, putting a reference would be a better choice.

Comment: In line 209-211, you said, “I re-code the randomized candidate issue positions to 210 indicate whether the candidate agrees with the respondent (1) or 211 not (0). ”. What do you mean by re-code? Clarify a little bit. Did you mean post-coding?

Comment: In line 211-212, you said “These agreement variables are the key independent 212 variables (see “Analysis” for more on methodology)”, here you are referring to something you described later. Usually reference is made to something described earlier. However, a little more is required to understand the concept of “agreement variables”. You didn’t mention this terminology earlier.

Comment: Then in line 213-216, you said “For the heterogeneous effects analysis, I make use of the 214 Common Content’s battery of demographic and political questions, 215 collapsing some categories to achieve adequate subgroup sample 216 sizes.”. Reference is needed here to justify that whatever you are doing scientifically appropriate.

Comment: The authors should indicate whether the study adhered to established ethical standards (e.g., the Declaration of Helsinki). Although the manuscript states that the study was approved by the Institutional Review Board (IRB), it would be helpful to explicitly mention compliance with relevant ethical guidelines. If the research complied with these standards, please state this clearly in the Ethics section.

Comment: In line 236-237, you mentioned “I follow the analytical strategy of Alvarez and Morrier with a 237 slight modification. ”. Again you are referring to something, need to put a reference.

Comment: Do you have explicit reason behind using 90% CI. Usually 95% is used. If you are using 90%, you made concrete justification for that.

Comment: In the discussion and conclusion you mention relatively small sample size as one of you limitations. However, 1000 participant is not a small sample.

Comment: Appendix S2. The frequency table is not representative. If you could add percentage along with the frequency, it would better represent.

Comment: Appendix S4 is a bit confusing. Try to incorporate more information about the items of table in the suffix.

**Do you want your identity to be public for this peer review?** For information about this choice, including consent withdrawal, please see our Privacy Policy

Reviewer #1: No

Reviewer #2: **Yes:** Yonas Angaw

Reviewer #3: No

---

## [Author Response · Author response to Decision Letter 1]

13 Jan 2026

Please see uploaded revision letter.

---

## [Decision Letter · Decision Letter 1]

25 Jan 2026

Who cares about mental health? Benchmarking the issue importance of mental health

for American voters

PONE-D-25-40424R1

Dear Dr. <!--StartFragmentHaselswerdt<!--EndFragment,

We’re pleased to inform you that your manuscript has been judged scientifically suitable for publication and will be formally accepted for publication once it meets all outstanding technical requirements.

Kind regards,

Omar El Deeb

Academic Editor

PLOS One

Additional Editor Comments (optional):

Reviewers' comments:

Reviewer's Responses to Questions

**Comments to the Author**

Reviewer #1: All comments have been addressed

2. Is the manuscript technically sound, and do the data support the conclusions?

Reviewer #1: Yes

3. Has the statistical analysis been performed appropriately and rigorously?

Reviewer #1: Yes

4. Have the authors made all data underlying the findings in their manuscript fully available?

Reviewer #1: Yes

5. Is the manuscript presented in an intelligible fashion and written in standard English?

Reviewer #1: Yes

Reviewer #1: Thank you for addressing my feedback. I have no further comment. The author did what they could. I wish them my best wishes for their paper.

**Do you want your identity to be public for this peer review?** For information about this choice, including consent withdrawal, please see our Privacy Policy

Reviewer #1: No

---

## [Editor Report · Acceptance letter]

PONE-D-25-40424R1

PLOS One

Dear Dr. Haselswerdt,

I'm pleased to inform you that your manuscript has been deemed suitable for publication in PLOS One. Congratulations! Your manuscript is now being handed over to our production team.

Kind regards,

on behalf of

Dr. Omar El Deeb

Academic Editor

PLOS One